# Screening of Thymoquinone Content in Commercial *Nigella sativa* Products to Identify a Promising and Safe Study Medication

**DOI:** 10.3390/nu14173501

**Published:** 2022-08-25

**Authors:** Elisabeth Khaikin, Sigrun Chrubasik-Hausmann, Sebahat Kaya, Benno F. Zimmermann

**Affiliations:** 1Institute of Nutritional and Food Sciences, Food Sciences, University of Bonn, Meckenheimer Allee 166a, 53115 Bonn, Germany; 2Institute of Forensic Medicine, University of Freiburg, Albertstr. 9, 79104 Freiburg, Germany; 3Department of Oral and Maxillofacial Surgery Plastic Surgery, University of Mainz, Augustusplatz 2, 55131 Mainz, Germany

**Keywords:** thymoquinone, *Nigella sativa*, black seed oil, HPLC, food supplement

## Abstract

(1) Background: Thymoquinone (TQ) is the leading compound accounting for the pharmacological effects of *Nigella sativa* seed oil, also known as black seed oil. This study aimed to analyze the TQ content of commercial black seed oils and black seed oil-containing capsules to obtain information on the quality of the products and to find a promising and safe study medication for a putative clinical study. (2) Methods: Six black seed oils and five black seed oil-containing capsules were analyzed. TQ was quantified using a validated method consisting of a simple methanolic extraction and a fast HPLC-UV analysis. (3) Results: The TQ content varied from 3.08 to 809.4 mg/100 g (mean). The highest TQ content was found in a bottled oil, which might be considered for a clinical study. A dose of 4 mL of this oil per day contains 30 mg TQ, which is unlikely to be harmful. Based on the literature, a safe daily TQ dosage appears to be <48.6 mg per adult. (4) Conclusions: These findings suggest that black seed products should be regulated regarding TQ content to enable consumers to buy black seed food supplements of known content for the maintenance and improvement of health.

## 1. Introduction

Since ancient times, the seeds of *Nigella sativa* (black seed) and its oil have been appreciated for their preventive and curative health benefits. Evidence of effectiveness has been investigated in relation to metabolic conditions such as diabetes mellitus, hypertension, and hypercholesterolemia [1]. However, the number of studies suggesting its efficacy for treating other conditions is continuously increasing [2]. Thymoquinone (TQ) is the leading compound involved in black seed mechanisms of action. A previous study analyzing 10 black seed oil products marketed in Malaysia identified a 27-fold difference in the TQ content in bottled oil and oil capsules. The authors recommended that the TQ content in black seed commercial oil products should be regulated, perhaps even by fortifying oils with standard TQ [3]. The authors did not suggest a TQ concentration in black seed oil or a daily dose to be applied.

Factors affecting the TQ content in black seed oil products include the geographical area of black seed cultivation [4], the time of harvesting the seeds [5], the duration and condition of seed storage prior to the manufacturing process, and the manufacturing process itself [6]. The TQ content in the seeds increased steadily toward the end of seed maturation on day 75 after fertilization, as did the amounts of other coactive ingredients, such as p-cymene, thymohydroquinone, α-thujene and carvacrol, and the minor components α-pinene, sabinene, and β-pinene [5]. Although TQ and its derivatives are the contributors to the antioxidative effect of black seed oils, the thymol isomers carvacrol, t-anethole, and 4-terpineol contribute to the overall antioxidative activity [7].

To maximize the shelf life of black seed purchased in bulk and to better retain flavor and potency, the seed should be stored in containers with tight-fitting lids. Determinants that affect the stability of the volatiles include air, heat, and light [8].

Cold press extraction is the conventional method to obtain oil from black seeds. It involves no heat and/or chemicals and is therefore preferred by consumers. However, cold press extraction provides low yields and is more susceptible to oxidation than solvent-extracted oils [9]. Novel pretreatment techniques, such as microwaving, enzymatic digestion, pulsed electric field, and ultrasonication, have been introduced. These methods not only improve the oil yield and quality attributes but also reduce seed oil extraction time, solvents, and energy consumption. Higher phenolic compounds, carotenoids, tocopherols, phytosterols, and antioxidant properties in oil from pretreated seeds offer health benefits related to the prevention of cancer, diabetes, obesity, and inflammatory and cardiovascular diseases [6]. Applying supercritical fluid extraction can also increase the TQ yield compared to cold press extraction (6.4 mg/mL vs. 1.8 mg/mL) [10].

The aim of this study was to determine the TQ content of black seed oils and black seed oil-containing capsules available as food or food supplements on the Swiss market to obtain information on the quality of these products regarding possible health effects. Furthermore, we aimed to identify a black seed oil commercial product that contained sufficient TQ in the daily dose and could safely be employed in a putative clinical study.

## 2. Materials and Methods

### 2.1. Samples

Details of the samples are presented in Table 1. All samples were given by the Swiss distributors apart from sample 9, which was ordered online from Australia. According to the manufacturer, oil sample 6 was used to fill the capsule of sample 11. All products were analyzed before the best-before date was reached with the exception of sample 8: the best-before date of this sample had been exceeded by six months at the date of analysis.

### 2.2. Chemicals

Acetonitrile, HPLC–MS grade (VWR International, Darmstadt, Germany), methanol, analytical reagent grade (Fisher Scientific, Loughborough, UK), water, LC–MS grade (Fisher Scientific, Schwerte, Germany), thymoquinone, analytical standard, 99.9% (Supelco, Sigma Aldrich, Merck KGaA, Darmstadt, Germany).

### 2.3. Analysis of Thymoquinone

#### 2.3.1. Extraction

Thymoquinone (TQ) was extracted according to the method of Soliman et al. [11] with slight modifications, i.e., 80–150 mg of the oil samples was weighed into a volumetric flask of 10 mL. In the case of capsules, ten capsules were pierced by a needle, and the oily content was pressed out into a small beaker and stirred. A total of 60–500 mg of the capsule content was weighed into a volumetric flask of 10 mL. In both cases, the volume was made up to 10 mL with methanol, and the flask was vortexed for 1 min. The oil was allowed to reaggregate (which took less than a minute), and approximately one milliliter of the methanolic phase was withdrawn by a single-use syringe and pushed through a membrane filter (Chromafil PET, 20/15 MS, pore size 0.2 µm; Macherey-Nagel, Düren, Germany) into the HPLC vial. All extraction steps were executed by avoiding direct sunlight in amber glassware or wrapped with aluminum foil.

#### 2.3.2. HPLC Analysis

A 2 µL aliquot of the methanolic extract was injected into the HPLC (Agilent 1100 with degasser G1379A, binary pump G1312A, autosampler G1313A, column oven G1316A and variable wavelength detector G1314A; Agilent, Santa Clara, CA, USA). Analysis was based on the method by Alkhatib et al. [3] with some modifications, i.e., the eluent was acetonitrile + water (60 + 40, v + v; isocratic) at a flow rate of 0.8 mL/min. The column (XSelect CSH C18, 3 mm × 150 mm, 3.5 μm with a guard column of the same material, 2.1 mm × 5 mm; Waters, Eschborn, Germany) was maintained at 40 °C. The resulting backpressure was approximately 220 bars. Chromatograms were recorded at 254 nm, i.e., the maximum of the TQ UV spectrum [12,13]. Data were recorded for 10 min after each injection, and peaks were integrated using the software PeakSimple (version 4.51, SRI Inc., Bad Honnef, Germany) via a PeakSimple Interface Box 202 (Schambeck SFD GmbH, Bad Honnef, Germany).

The TQ peak was identified by comparing the retention time and UV spectrum with an authentic reference compound. The retention time of TQ was 2.30 min. The chromatograms of most samples had two smaller but considerable peaks at 5.87 min and 8.03 min. That is why a total runtime of 10 min was necessary.

For quantification, an external calibration was performed: a stock solution of 2.7 mg TQ in 10 mL acetonitrile + water (60 + 40, v + v) was diluted with the same mix to eight solutions between 2.7 and 81 mg/L. The calibration curve was calculated as peak area = 5.4165 × concentration (mg/L) + 0.9429 with a coefficient of determination R^2^ of 0.9984.

All samples were extracted and analyzed in duplicate, except samples 2 and 11, which were analyzed in decuplicate in the context of method validation (see the following paragraph).

#### 2.3.3. Method Validation

Intraday repeatability was calculated as the relative standard deviation of triplicate or quadruplicate analysis of samples 2 (oil) and 11 (capsule) on three days. Interday repeatability was calculated as relative standard deviations of the means of the ten analyses on the three different days.

## 3. Results and Discussion

### 3.1. Analytical Method

Extraction was simple using a single extraction step with methanol. Recoveries of TQ from oil samples extracted by methanol were 100.0% ± 1.1% and 100% ± 2.5%, as shown by Soliman et al. [11] and by Aboul-Enein et al. [14], respectively. Other authors applied more complicated methods, including solvent evaporation after extraction [14,15] or solid phase extraction [16]. These steps seem to be unnecessary providing that the detection is sufficiently sensitive.

With a total duration of 10 min, this chromatographic method is as fast as the fastest HPLC methods for TQ published thus far [3,14] apart from one faster method of only six minutes runtime using UHPLC [15]. The linearity of the calibration function was tested in the concentration range of 2.7 to 81 mg/L. The coefficient of determination R^2^ was calculated as 0.9984.

The intraday and interday repeatability of the entire analytical process used here were determined by analyzing an oil sample and a capsule sample. The results can be found in Table 2. In the same publications as mentioned above [3,14,15], repeatability was determined using standard solutions and are, therefore, hardly comparable to the values found here. Other reports found similar values for the precision of their analytical HPLC methods for TQ [16,17].

### 3.2. TQ Content

The content of TQ in the samples varied considerably (Table 3) and ranged from 3 to 809 mg/100 g. TQ concentrations in black seed oil published thus far varied even more. Alkhatib et al. [3], who analyzed five different black seed oil samples, found TQ amounts between 140 and 1880 mg/100 g, which was comparable to the results presented here. The results reported by Aboul-Enein et al. [14] and Ghosheh et al. [16] were somewhat lower but still of the same order of magnitude (33 and 132 mg/100 mL; 53 mg/100 g, respectively). In one black seed oil in the study by Aboul-Enein et al. [14], TQ was not detected and TQ was extremely low (0.0071 mg TQ/100 mL) in the oil sample of another study [18]. The latter group had previously found 7.2 and 8.2 mg TQ/100 mL in other samples [11].

In the capsules analyzed here, the TQ content varied slightly more (21.5 to 661 mg/100 g, difference by a factor of approximately 31; excluding the expired sample 8) compared to the bottled oils (45.3 to 809 mg/100 g, difference by a factor of almost 18). Capsule sample 8 provided by far the lowest TQ content and the highest standard deviation. The reason for this might be that the best-before date of this sample had been exceeded by six months at the date of analysis. According to the manufacturer, sample 6 was used to fill the capsule of sample 11. The TQ concentration in the oil/capsule pair was rather similar.

The reasons for the wide range of the TQ content in black seed oils may be the variations of the TQ content in the seeds, which likely depends on the geographical area of the black seed cultivation [4] and the time of harvesting the seed [5]. Seed storage and the extraction process have a significant impact on the amount of TQ in the oil [6]. The TQ content of the fresh oil may decrease during storage due to heat, light, and oxygen exposure [8,9,11,15].

Interestingly, all cited publications present data from Egyptian black seed oils analyzed in Egypt. In this study, all black seed oils originated from Egypt, apart from one sample from India. However, the capsules were produced in Switzerland and Australia and are available in Switzerland and Australia, respectively.

### 3.3. Safety and Efficacy

The *N. sativa* seeds and the oil made thereof are generally recognized as safe (GRAS) by the United States Food and Drug Administration (USFDA) (Code of federal regulation: 21CFR182.10) [19]. In two clinical studies, in which 5 mL/day of black seed oil was administered over eight weeks, no adverse or toxic effects were observed [20,21]. Supplementation with black seed oil at doses of 1.5, 3.0, and 4.5 mL/day for 20 days was also well tolerated and did not affect the number of blood cells, hepatic and renal variables, or immune response in healthy subjects [22]. Unfortunately, the TQ content of the oils in these studies was not stated.

In animal studies, the acute oral LD_50_ of TQ was 794 mg/kg in rats and 871 mg/kg in mice, which is 100 to 150 times higher than the TQ doses needed to produce anti-inflammatory, antioxidant, and anticancer effects [23]. However, chronic consumption of black seed oil over longer periods may be harmful. When rats were given 1 mL [24] or 2 mL [25] *N. sativa* oil per kg over 12 weeks, an increase in hemoglobin and hematocrit, and a decrease in the number of leukocytes and platelets occurred. A slowing of body weight gain was also observed in black seed-treated rats compared to control animals. Oral doses of black seed oil of 15 and 25 mL/kg over four weeks induced changes in the histological structure of the renal cortex and, to a lesser extent, of the liver cells [26]. The subchronic (90-day) repeated dose toxicity study in Wistar rats according to the OECD guidelines revealed a TQ NOAEL (no adverse effect level) of 5 mg/kg. Upon conversion to human dosage, the safe daily TQ dosage was calculated to be below 48.6 mg per adult [27]. In a Phase-I-Study, the safe use of 200 mg black seed oil formulation with 5% TQ (10 mg TQ in the daily dosage) was confirmed in 35 healthy volunteers over 90 days versus placebo (n = 35) [28]. Based on the data available, we suggest that sample 5 at a dose of 3.7 g or 4 mL (supposing a specific mass of 0.92 g/mL) per day containing 30 mg TQ may safely be employed in a putative clinical study over six weeks.

## 4. Conclusions

The results of our study suggest that the thymoquinone content should be declared on all black seed oil products to enable researchers to choose a suitable study medication and consumers to buy black seed food supplements that contain sufficient active principle for the maintenance and improvement of health.

## Figures and Tables

**Table 1 nutrients-14-03501-t001:** Details of the analyzed samples as stated on the packages.

Sample	Product	Origin	Lot Number and Best before	Further Ingredients	Suggested Dose	Amount of Liquid in One Capsule
1	Oil	Egypt	1000016256 6.1.24	None	Not stated	—
2	Oil	Egypt	984082_2000_EG 25.2.24	None	Not stated	—
3	Oil	Egypt	0000098540 31.07.2024	None	Not stated	—
4	Oil	Egypt	202107.0179 30.10.22	None	Not stated	—
5	Oil	Egypt	E18BS 27.1.23	None	Not stated	—
6	Oil	Egypt	2101791 31.01.2025	Rosemary Extract, Tocopherols	Not stated	—
7	Oil in Capsules	Egypt	2100676 30.6.24	Vitamin E, Rosemary Extract	3 Capsules per Day	500 mg
8	Oil in Capsules	Egypt	1900425 31.10.2021	Rosemary Extract, Tocopherols	Not stated	500 mg
9	Oil in Capsules	India	28945A 3.11.23	None	2 Capsules per Day	500 mg
10	Oil in Capsules	Egypt	980377_1700_EG 6.7.24	Natural Vitamin E, modified	6 Capsules per Day	760 mg
11	Oil in Capsules	Egypt	2102609 30.4.25	Vitamin E, Rosemary Extract	3 Capsules per dDy	500 mg

**Table 2 nutrients-14-03501-t002:** Method validation data.

Linearity					
Concentration range: 2.7–81 mg/L			Coefficient of determination R^2^: 0.9984		
**Recovery** ± standard deviation			Reference		
100.0% ± 1.1%			Soliman et al. [11]		
100% ± 2.5%			Aboul-Enein et al. [14]		
**Precision**					
Sample	Day	n	Mean Thymoquinone Content in mg/100 g	Intraday Relative Standard Deviation	Interday Relative Standard Deviation
2 (oil)	1	3	544.4	0.87%	1.26%
	2	3	531.0	1.36%	
	3	4	539.1	0.48%	
11 (capsules)	1	3	45.1	1.88%	7.54%
	2	3	44.0	5.83%	
	3	4	39.0	2.87%	

n = number of analyses.

**Table 3 nutrients-14-03501-t003:** Thymoquinone contents.

Sample	Thymoquinone in mg/100 g (Mean)	SD	rel. SD		
Oil Samples					
1	133.9	0.24	0.18%		
2	538.2	7.02	1.30%		
3	172.4	0.90	0.52%		
4	138.9	1.25	0.90%		
5	809.4	2.69	0.33%		
6	45.26	4.67	10.3%		
Capsule Samples				Recommended Daily Dose in mg	TQ in Daily Dose in mg
7	27.75	0.54	1.94%	1500	0.416
8	3.08	0.88	28.7%	—	—
9	660.5	0.88	0.13%	1000	6.61
10	21.46	3.59	16.7%	4560	0.979
11	42.33	3.21	7.58%	1500	0.635

SD = standard deviation; rel. SD: relative standard deviation; these are the standard deviations of the analytical results of the decuplicates (samples 2 and 11) or duplicates (all other samples).

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
