# Peer review of "Screening of Thymoquinone Content in Commercial Nigella sativa Products to Identify a Promising and Safe Study Medication"

_nutrients, 2022, doi:10.3390/nu14173501_

Round 1

Reviewer 1 Report

The content of Thymoquinone in each capsule is varied (table 3). Some pills had been exceeded by six months. 

Authors should present the safety data in a table or figures.

The authors showed the Thymoquinone content (table 3) in each sample; however, why are there SD or rel SD for "each" piece? 

Author Response

The content of Thymoquinone in each capsule is varied (table 3). Some pills had been exceeded by six months. 

Answer: all products were analyzed, before the best-before date was reached with the exception of sample 8: the best-before date of this sample had been exceeded by six months at the date of analysis as mentioned in section 3.2. For the sake of clarity, this information has now been added in section 2.1.

Authors should present the safety data in a table or figures.

Answer: there is a very limited number of published data on the safety of TQ. We have summarized them in section 3.3. Since reviewer 2 did not claim on that, we would prefer to leave the section as it is.

The authors showed the Thymoquinone content (table 3) in each sample; however, why are there SD or rel SD for "each" piece? 

Answer: The SD and relative SD in table 3 are the standard deviation of the analytical results, i.e., the duplicates and in case of the samples 2 and 11 the decuplicates. For the sake of clarity, this information has been added in table 3.

Reviewer 2 Report

in this paper it was a question to help determine the content of TQ (main Bioactive metabolite) in the oil of the nigelle, the contents of 11 commercial samples were determined by HPLC-UV, however, the validation of the quantification method as well as the discussion are incompletes probably the authors have not integrated all their results. Apart from assessing intra and interday accuracy, nothing else has been performed for the method validation.

There is another table 2 at the end of the manuscript, but I ignore what it state.

reference 1 is incomplete...

Author Response

in this paper it was a question to help determine the content of TQ (main Bioactive metabolite) in the oil of the nigelle, the contents of 11 commercial samples were determined by HPLC-UV, however, the validation of the quantification method as well as the discussion are incompletesprobably the authors have not integrated all their results. Apart from assessing intra and interday accuracy, nothing else has been performed for the method validation.

Answer: we did integrate all our results, but obviously they were not clearly presented. In addition to the intraday and interday accuracy, also recovery or accuracy is known, which had been determined earlier. This was mentioned in the beginning of section 3.1. just shortly. Also, linearity was proven in a certain range of concentrations and was mentioned in section 2.3.1. For the sake of clarity, the recovery and linearity values have now been added in section 3.1 and in table 2.

There is another table 2 at the end of the manuscript, but I ignore what it state. 

Answer: we forgot to delete a dummy table of the template. Sorry!

reference 1 is incomplete...

Answer: we have completed reference 1.

Round 2

Reviewer 2 Report

In my opinion the paper can now be accepted as a short report but not as a full length Research article in Nutrients, indeed the work only reports quantification of TQ in different batch of nigella sativa oils using HPLC/VWD and nothing else